# *Calocedrus formosana* Essential Oils Induce ROS-Mediated Autophagy and Apoptosis by Targeting SIRT1 in Colon Cancer Cells

**DOI:** 10.3390/antiox13030284

**Published:** 2024-02-26

**Authors:** Atikul Islam, Yu-Chun Chang, Nai-Wen Tsao, Sheng-Yang Wang, Pin Ju Chueh

**Affiliations:** 1Institute of Biomedical Sciences, National Chung Hsing University, Taichung 40227, Taiwan; d107059008@mail.nchu.edu.tw (A.I.); yujun0715@dragon.nchu.edu.tw (Y.-C.C.); 2Special Crop and Metabolome Discipline Cluster, Academy Circle Economy, National Chung Hsing University, Taichung City 402202, Taiwan; nwt1228@dragon.nchu.edu.tw; 3Department of Forestry, National Chung Hsing University, Taichung 40402, Taiwan; taiwanfir@dragon.nchu.edu.tw; 4Graduate Institute of Basic Medicine, China Medical University, Taichung 40402, Taiwan; 5Department of Medical Research, China Medical University Hospital, Taichung 40402, Taiwan

**Keywords:** colorectal cancer, *Calocedrus formosana*, essential oils, reactive oxygen species (ROS), autophagy, apoptosis, silent mating type information regulation 2 homolog 1 (SIRT1)

## Abstract

Colorectal cancer is the most common cancer that affects both sexes and has a poor prognosis due to aggressiveness and chemoresistance. Essential oils isolated from *Calocedrus formosana* (CF-EOs) have been shown to demonstrate anti-termite, antifungal, anti-mosquito, and anti-microbial activities. However, the anticancer effects of CF-EOs are not yet fully understood. Therefore, the present study aimed to explore the molecular mechanism underlying CF-EOs-mediated anti-proliferative activity in colon cancer cells. Here, cell impedance measurements showed that CF-EOs inhibit proliferation in colon cancer cells with wild-type or mutant p53. Flow cytometry revealed that CF-EOs at 20, 50 µg/mL significantly induced ROS generation and autophagy in both HCT116 p53-wt and HCT116 p53-null cell lines, whereas pretreatment with the ROS scavenger N-acetyl cysteine (NAC) markedly attenuated these changes. CF-EOs also induced apoptosis at 50 µg/mL in both lines, as determined by flow cytometry. Protein analysis showed that CF-EOs markedly induced apoptosis markers, including Trail, cleaved caspase-3, cleaved caspase-9, and cleaved PARP, as well as autophagy markers, such as the levels of ULK1, Atg5, Atg6, Atg7, and the conversion of LC3-I to LC3-II. CF-EOs were further found to inhibit the activity and expression of the NAD^+^-dependent deacetylase SIRT1 to increase the levels of acetylated p53 (Ac-p53) in p53-wt cells and acetylated c-Myc (Ac-c-Myc) in p53-null cells, ultimately inducing apoptosis in both lines. Interestingly, suppression of SIRT1 by CF-EOs enhanced the acetylation of ULK1, which in turn prompted ROS-dependent autophagy in colon cancer cells. The induction of apoptosis and autophagy by CF-EOs suggests that they may have potential as a promising new approach for treating cancer. Collectively, our results suggest that essential oils isolated from *Calocedrus formosana* act as a promising anticancer agent against colon cancer cells by targeting SIRT1 to induce ROS-mediated autophagy and apoptosis.

## 1. Introduction

According to the World Health Organization, cancer is the foremost cause of death worldwide, accounting for close to 10 million deaths in 2020. Such numbers include a great many colon cancer cases, with an estimated 106,180 new cases and 52,580 deaths reported in the United States in 2022. The extended trends in universal cancer constancy rates indicate that colon cancer rates decreased overall by 2% per year from 2014 to 2018, but those in adults younger than 50 years increased by 1.5% per year [1]. The incidence of colorectal cancer (CRC) is greatly increasing in Asia and Eastern European countries. Advances in treatment options for CRC have been optimized to attain the best results with low morbidity [2], and various therapeutic options have been reported, such as combination therapy, anti-EGFR agents, and anti-angiogenic agents. However, the efficacy of such treatments remains limited by chemotherapeutic resistance and high cancer-related mortality [3,4]. The lack of complete remission, especially among patients with advanced-stage CRC, further confines the effectiveness of therapeutic drugs aimed at managing CRC.

Of late, natural compounds with anticancer properties have received much attention through extensive anticancer drug discovery and screening programs approved by the US National Cancer Institute (NCI) [5]. At present, many of them are used in cancer therapy that are originally from natural sources, and some of them are isolated from natural products [6]. Intriguingly, studies have shown that natural compounds attack cancer cells, halt tumor eruption and increase the sensitivity of chemotherapeutic compounds to various cancer cell lines including breast, liver, bladder, colorectal, and gastric cancer [7,8,9,10]. 

Apoptosis and autophagy are two distinct cellular processes that play vital roles in maintaining cellular homeostasis and cell growth control, including cancer cells [11,12,13]. Autophagy is a process of cellular self-digestion, while apoptosis is a process of self-destructive cell death. Both processes play important roles in preventing and treating cancer [14]. Essential oils are a mix of plant volatiles consisting of secondary metabolites that gained popularity for their wide-ranging utility, especially in fragrances and skincare products [15,16]. Importantly, *Calocedrus formosana* (syn. *C*. *macrolepis* var. *formosana*), also known as Taiwan incense cedar, has been reported to exert various biological activities, such as anti-bacterial and antifungal [17], anti-termitic [18], and anti-melanogenic [19]. Moreover, carvacrol, a leading component of oregano and thyme essential oils, was shown to inhibit cell proliferation and migration in HCT116 cells [20]. Essential oils derived from *Citrus aurantifolia*, *Satureja montana*, and *Carum copticum* reportedly have anti-proliferative activity and induce apoptosis against numerous cancer cell lines [21,22,23,24,25,26]. Other studies have shown that the leaf extracts of *C. Formosana,* florin, inhibit cell growth and induce apoptosis in bladder cancer cells [27], and those of yatein and hinokitiol induce ROS production and apoptosis and inhibit cell migration in non-small-cell lung cancer cells [28,29]. Traditional medicinal herbs, such as pogostone from *Pogostemon cablin* and eugenol (a phenolic phytochemical) extracted from certain essential oils, have been shown to provoke apoptosis and autophagy in human colorectal and breast cancer cells [30,31]. Additionally, amurensin G isolated from grape reportedly inhibits SIRT1 and enhances the apoptotic effects of TNF-related apoptosis-inducing ligand (TRAIL) in TRAIL-resistant K562 cells [32]. Curcumin has also been shown to inhibit the activity and expression of SIRT1 and reduce the invasiveness of colon cancer cells [33]. All these lines of evidence suggest that essential oils have the potential for use as phytomedicines.

ROS are highly reactive molecules that are produced in cells in response to external stimuli or stress. Under normal conditions, ROS remains at low levels and is tightly controlled and plays an important role in cellular signaling and homeostasis. On the other hand, cancer cells exhibit higher basal levels of ROS that act a dual role in cancer metabolism [34]. Specifically, high levels of ROS trigger damage to macromolecules, such as DNA and proteins, resulting in cell death [35,36]. This phenomenon has been observed in cancer cells treated with EOs. For example, in a previous study, treatment with EOs from *Aniba rosaeodora* (rosewood) was found to induce apoptosis in human epidermoid carcinoma A431 and immortalized HaCaT cells through ROS generation [35]. Similar effects were observed with EOs of *Zanthoxylum schinifolium* in liver cancer cells (HepG2) [37]. However, the role of ROS in colon cancer cells with CF-EOs has not yet been fully explored.

p53 is known as the “guardian of the genome”, acting as a tumor suppressor. Studies have shown that p53-mediated apoptosis is important in the prevention of tumor development [38]. However, the role of p53 in autophagy is more complex and paradoxical, depending on its subcellular location [39,40,41]. Given that p53 is often mutated or inactivated in CRC patients, in the present study, we examined the effects of *C. formosana* isolated essential oils (CF-EOs) utilizing two colon cancer cell lines with different p53 functionality.

## 2. Materials and Methods

### 2.1. Materials

The anti-SIRT1, anti-Atg5, anti-ULK1, anti-PARP, anti-TRAIL, anti-c-Myc, anti-p53, anti-acetyl-p53, anti-acetylated lysine, anti-cleaved caspase-3, and anti-caspase-9 antibodies were purchased from Cell Signaling Technology, Inc. (Beverly, MA, USA). The anti-β-actin and anti-acetyl-c-Myc antibodies were from Millipore Corp. (Temecula, CA, USA). The anti-Beclin 1 (Atg 6), anti-Atg7, and anti-LC3 antibodies were obtained from Novus Biologicals (Centennial, CO, USA). All primary antibodies were diluted according to the manufacturer’s recommendation. The anti-mouse and anti-rabbit IgG antibodies were purchased from Jackson Immuno Research Laboratories INC. and other reagents, namely, MG132 and Bafilomycin A1 (Baf-A1), were purchased from MedChemExpress (SouthBrunswick Township, NJ, USA) and Biosynth-Carbosynth (FB146475) (Compton, UK) and dissolved in DMSO from Sigma Chemical Company (St. Louis, MO, USA) unless otherwise specified.

### 2.2. Cell Culture

HCT116 (human colorectal cancer) p53-wild-type (Bioresource Collection and Research Center, BCRC, Hsinchu, Taiwan), p53-null (Horizon Discovery, Cambridge, UK), and BSEA-2B (Lung epithelial cell) cells were grown in McCoy’s 5A, RPMI, and RPMI medium, respectively, supplemented with 10% FBS, 100 units/mL penicillin, and 50 µg/mL streptomycin. Cells were maintained at 37 °C in a humidified atmosphere containing 5% CO_2_, and the media were replaced every 2–3 days. Cells were treated with different concentrations of essential oils isolated from *Calocedrus formosana* (dissolved in DMSO).

### 2.3. Preparation of C. formosana Wood Essential Oils (CF-EOs)

*C. formosana*, a cut boot from a 35-year-old tree, was collected in Nantou County, Taiwan in June 2019 that was identified by Professor Sheng-Yang Wang from the Department of Forestry of National Chung Hsing University. The voucher specimen (Y.S. Tseng 4325, TCF) was placed in the herbarium of the same university. A total of 400 g of air-dried wood crisps and 1.6 L (ratio 1:4) of water were hydro-distilled in a Clevenger apparatus for 6 h and the oil content was then determined. The average distillation rate of essential oils was approximately 1.34 g per hour. The essential oils of the wood were stored in airtight sample jars before analysis by gas chromatography–mass spectrometry (GC–MS) and bioactivity evaluation.

### 2.4. Gas Chromatography–Mass Spectrometry (GC-MS) Analysis

GC-MS was used to analyze the chemical composition of the CF-EOs using an ITQ 900 mass spectrometer in conjunction with a DB-5MS column. The temperature of the GC oven was initially set at 40 °C for 3 min, then increased to 180 °C at a rate of 3 °C/min and then increased to 20 °C /min until 280 °C was reached and held for 5 min. The injection temperature was maintained at 240 °C, while the ion source temperature remained at 200 °C during the analysis. Helium was used as the carrier gas at a rate of 1 mL/min, and the mass scanning range was 40–600 *m*/*z*. Standard databases such as Wiley/NBS Registry of mass spectral databases (V. 8.0, Hoboken, NJ, USA) and the National Institute of Standards and Technology (NIST) Ver. 2.0 GC/MS libraries were used to identify the volatile compounds. The Kovats index (KI) was calculated from the retention times of a compound using a homologous series of n-alkanes C9–C24. Standards were co-injected to identify the main components.

### 2.5. Identification of Unknown Compounds

The three unique CF-EOs compounds such as shonanic acid, thujic acid, and chaminic acid were purified and identified by chromatography and spectroscopy techniques as described previously [19]. In brief, using the Agilent 1100 HPLC system with UV detector, the CF-EOs were further separated. For the establishment of COSMOSIL C18-AR-II (10 mm I.D. × 250 mm, Nacalai Tesque Inc., San Diego, CA, USA), the solvent system MeOH/H_2_O was used. The gradient elution profile at a flow rate of 1.6 mL/min was as follows: 0 to 3 min, MeOH:H_2_O = 55:45; 3 to 13 min, MeOH:H_2_O = 75:25; 13 to 18 min, MeOH:H_2_O = 75:25; 13 to 18 min, MeOH:H_2_O = 75:25; 18 to 25 min, MeOH:H_2_O = 85:15; 25 to 30 min, MeOH:H_2_O = 85:15; 30 to 33 min, MeOH:H_2_O = 100:0; 5; 33 to 40 min, MeOH:H_2_O = 100:0. The wavelength of the UV detector was set to 254 nm. To identify the structure, the compounds were dissolved in deuterated chloroform (d-chloroform; CDCl3). 1H, 13C, and 2D NMR spectra were recorded on a Bruker AVANCE III NMR spectrometer (Bruker, Billerica, MA, USA), with 1H data acquired at 400 MHz and 13C data acquired at 100 MHz using standard experiments from the Bruker pulse program library. High-resolution mass spectrometry (HRMS) was determined using an LTQ Orbitrap XL (Thermo Fisher Scientific, Waltham, MA, USA). The detected compounds were evaluated by MS and 400 MHz NMR.

### 2.6. Continuous Observation of Cell Proliferation by Cell Impedance Determination

For uniform observation of changes in cell proliferation, 10^4^ cells/well of HCT116 p53-wt and p53-null and 5 × 10^3^ cells/well of BSEA-2B were plated onto E-plates and incubated for 30 min at room temperature and then placed onto the xCELLigence System (Roche, Mannheim, Germany). Cells were grown overnight before being treated with different concentrations of CF-EOs and impedance was measured every hour. Cell impedance is defined by the cell index (CI) = (Z_i_ − Z_0_) [Ohm]/15[Ohm], where Z_0_ is background resistance and Z_i_ is the resistance at an individual time point. A normalized cell index was determined as the cell index at a certain time point (CI_ti_) divided by the cell index at the normalization time point (CI_nml_time_).

### 2.7. Cell Viability Assay

Cells (5 × 10^3^) were cultured in 96-well culture plates and allowed to adhere overnight at 37 °C in a growth medium containing 10% FBS. The cells were exposed to different concentrations of CF-EOs for 24 h, and after 24 h of incubation with different concentrations of CF-EOs, the cells were treated with a 0.5 mg/mL solution of 3-(4,5-dimethylthiazolyl-2)-2,5-diphenyltetrazolium bromide (MTT, 20 μL/well) for 3 h at 37 °C. The number of viable cells was determined by MTT uptake measured at 495 nm according to the manufacturer’s instructions. All experiments were performed in at least triplicate on three separate occasions.

### 2.8. Apoptosis Determination

An Annexin V-FITC Apoptosis Detection Kit (BD Pharmingen, San Jose, CA, USA) was utilized to measure apoptosis. Cells were cultured in 6 cm culture dishes, exposed to CF-EOs, trypsinized, and harvested by centrifugation. Each pellet was rinsed with PBS, resuspended in 1x binding buffer, and stained with Annexin V-FITC (fluorescein isothiocyanate) followed by propidium iodide (PI; to determine necrotic or late apoptotic cells). The percentages of viable (FITC-negative and PI-negative), early apoptotic (FITC-positive and PI-negative), late apoptotic (FITC-positive and PI-positive), and necrotic (FITC-negative and PI-positive) cells were evaluated by a Beckman Coulter FC500 flow cytometer and cytoFLEX. The results are expressed as a percentage of total cells.

### 2.9. Autophagy Determination

Autophagosomes, acidic intracellular compartments that mediate the degradation of cytoplasmic material during autophagy, were visualized by staining with acridine orange (AO). Cells were washed with PBS, stained with 2 mg/mL AO for 10 min at 37 °C, washed, trypsinized, and analyzed using a Beckman Coulter FC500. Results are expressed as a percentage of total cells.

### 2.10. Determination of Reactive Oxygen Species (ROS)

ROS were examined by determining the generation of intracellular hydrogen peroxide (H_2_O_2_) via 5-(6)-carboxy-2′,7′-dichlorodihydrofluorescein diacetate (carboxy-H2DCFDA-cellular ROS assay kit) (ab 113851) staining. In brief, cells (2 × 10^5^) were exposed to CF-EOs for a total of one hour, rinsed with PBS, and treated with 5 μM H2DCFDA in 1× binding buffer for 30 min. The cells were then harvested by trypsinization and centrifugation, rinsed with PBS, centrifuged at 11,000 rpm for 5 min, resuspended the pellet with PBS, and examined immediately by a Beckman Coulter FC500 flow cytometer.

### 2.11. Determination of SIRT1 Deacetylase Activity In Vitro

SIRT1 deacetylase activity was determined using a SIRT1 activity assay kit (fluorometric) (ab156065) according to the manufacturer’s protocol. Briefly, the fluorophore and quencher are coupled to the amino and carboxyl ends of the substrate peptide, respectively, and fluorescence cannot be emitted before the deacetylase reaction. However, when SIRT1 performs deacetylation, the substrate peptide is cut by the action of the simultaneously added protease, the quencher separates from the fluorophore, and fluorescence is emitted. Developer was added to each well of the microtiter plate and mixed well. The background was determined in wells containing 30 µL Assay Buffer and 5 µL DMSO. To measure the modulation of SIRT1 activity by CF-EOs, 5 µL of CF-EOs were added to wells containing 25 µL Assay Buffer and 5 µL diluted human recombinant SIRT1. Reactions were started by adding 15 µL of the substrate solution to each well. The plate was then covered and incubated on a shaker for 30 min at room temperature. Reactions were stopped by adding 50 μL of stop solution to each well. Plates were read at 1 to 2 min intervals for 30 to 60 min in a fluorometer using an excitation wavelength of 340–360 nm and an emission wavelength of 440–460 nm. The rate of reaction while the reaction velocity remained constant was measured and calculated.

### 2.12. Western Blot Analysis

Cell extracts were prepared in lysis buffer (20 mM Tris-HCl pH 7.4, 100 mM NaCl, 5 mM EDTA, 2 mM phenylmethylsulfonyl fluoride (PMSF), 10 ng/mL leupeptin, 10 μg/mL aprotinin). Volumes of extract containing equal amounts of proteins (30 µg) were resolved by SDS-PAGE and transferred to PVDF membranes (Schleicher & Schuell, Keene, NH, USA), and the membranes were blocked, washed, and probed with the indicated primary antibody overnight. The membranes were washed, incubated with horseradish peroxidase-conjugated secondary antibody for 1 h, and developed using enhanced chemiluminescence (ECL) reagents (Amersham Biosciences, Piscataway, NJ, USA) according to the manufacturer’s protocol.

For immunoprecipitation, HCT116 p-53 cells were harvested from 10 cm dishes and cell extracts were prepared in lysis buffer. For total cell lysates, 50 μL of supernatants were separated and then 1 mg of extracted protein was incubated with 20 μL of Protein G Agarose Beads (for rabbit antibodies) for 1 h at 4 °C in rotation for pre-clearing. The ULK1 antibody or control IgG was incubated onto beads in 500 μL of lysis buffer overnight in rotation at 4 °C. Beads were precipitated by centrifugation at 3000 rpm for 2 min at 4 °C and washed three times with lysis buffer, and samples were prepared for Western blotting analysis.

### 2.13. Statistical Analysis

All data are expressed as the means ±SEs of three independent experiments. All statistical analyses were performed using SigmaPlot 12.5 (Copyright 2003–2013 Systat Software Inc., San Jose, CA, USA) or IBM SPSS statistics version 20 (Chicago, IL, USA). The significance of differences between control and treatment groups was calculated using a one-way ANOVA followed by an appropriate post-hoc test such as LSD and *p* < 0.05 was considered to be statistically significant.

## 3. Results

### 3.1. Extraction and Chemical Composition of CF-EOs

CF-EOs were extracted from the heartwood of *C. formosana* by hydro-distillation, with a yield of 2.01% (*w*/*w*). The major chemical components of CF-EOs were identified by GC-MS analysis, and their relative contents are listed in Table 1. The Kovats index (KI) is a fruitful reference for the speculative identification of compounds by gas chromatography/mass spectrometry. KI is calculated based on the retention times of a compound, which are related to the neighboring n-alkanes. In total, twenty-four of twenty-seven compounds were identified from the complex GC chromatograms by comparison with standard databases, including Wiley/NBS and NIST. However, the three chemical constituents that could not be identified in the standard databases, such as shonanic acid, chaminic acid, and thujic acid, were further isolated and purified by column chromatography and high-performance liquid chromatography (HPLC) after nuclear magnetic resonance (NMR) and mass spectrometry (MS). The total of CF-EOs was 85.34%. The major components of CF-EOs were found to be shonanic acid (34.86%), thujic acid (10.2%), chaminic acid (10.06%), L-α-terpineol (5.92%), α-bourbonene (4.02%), myrtenoic acid (3.84%), 4-terpineol (3.2%), and β-cymene (2.12%), which together accounted for 74.22% of the content of CF-EOs.

### 3.2. CF-EOs Inhibit the Proliferation of HCT116 p53-wt and HCT116 p53-Null Cells

Anticancer drugs derived from natural products are currently used in clinical practice. For example, the widely used chemotherapy drug, paclitaxel, is derived from the Pacific yew tree, *Taxus brevifolia*. Essential oils contain a variety of volatile compounds, some of which have been shown to have anticancer properties. These compounds work in different ways to eliminate cancer cells or prevent them from growing and dividing. In this study, we first evaluated the anti-proliferative ability of essential oils from *Calocedrus formosana* (CF-EOs) on human colon cancer cells with differing p53 status. We treated both HCT116 wild-type (wt) and p53-null colon cancer cells with CF-EOs at concentrations of 10, 20, and 50 µg/mL. Dynamic cell impedance measurements showed that CF-EOs significantly (*p*-value) decreased the growth of both cell lines (Figure 1A). Interestingly, the treatments of CF-EOs exhibited no significant anti-proliferation effect on human non-tumorigenic epithelial cell line BSEA-2B from the human bronchial epithelium (Figure 1B). Consistent with the cell impedance results, an MTT cell viability assay showed that CF-EOs reduced the viability of colon cancer cells at concentrations of 20 and 50 µg/mL (Figure 1C).

### 3.3. CF-EOs Induce ROS-Dependent Autophagy in Colon Cancer Cells

We herein found that CF-EOs significantly induced an anti-proliferative effect at 20 and 50 µg/mL and next explored pathways associated with CF-EOs-induced cytotoxicity. Acridine orange (AO) was used to measure autophagy induction after CF-EO treatments in colon cancer cells. At a concentration of 20 and 50 µg/mL, CF-EOs markedly induced autophagy in HCT116 p53-wt and p53-null cells (Figure 2A), whereas pretreatment with the autophagy inhibitor Baf-A1 significantly reduced CF-EOs effects to induce autophagy (Figure 2B). CF-EOs-induced autophagy was accompanied by the upregulation of the autophagy markers ULK1, Atg5, Atg6, Atg7, and cleaved LC3-II in both cell lines (Figure 2C). To further confirm the autophagic pathway induced by CF-EOs, we performed an immunoprecipitation assay and validated an enhancement of acetylated ULK1, an indication for autophagy (Figure 2D).

ROS act as upstream signaling molecules in various pathways involved in regulating cell survival and/or cell death. Induction of ROS generation by anticancer agents can induce autophagy. Next, to examine whether CF-EOs induce ROS, we used H_2_DCFD- cellular ROS dye and found that CF-EOs significantly induced ROS generation at 20 and 50 µg/mL analyzed by flow cytometry (Figure 3A). Pretreatment with the ROS scavenger NAC markedly inhibited the ROS generation induced by CF-EOs in both cell lines (Figure 3B). Importantly, in both p53-wt and p53-null colon cancer cells, NAC pretreatment substantially impaired the autophagy induced by CF-EOs at 20 and 50 µg/mL (Figure 3C), suggesting that CF-EOs-induced autophagy is ROS-dependent. To investigate the role of ROS in autophagy flux, HCT116 p53-wt cells were cotreated with or without autophagosome lysosome binding inhibitor chloroquine (CQ), NAC, and 50 µg/mL CF-EOs. Concomitant treatment with CQ and 50 µg/mL CF-EO increased LC3-II protein levels to a greater extent than in the CF-EOs alone group, indicating an increased accumulation of autophagosomes and enhancement in autophagic flux. Surprisingly, concomitant treatment with the ROS inhibitor NAC markedly attenuated LC3-II expression induced by 50 µg/mL CF-EOs alone as well as by the combination of CQ and 50 µg/mL CF-EOs, suggesting that ROS is essential for autophagosomes and autophagy flux as analyzed by Western blotting (Figure 3D).

### 3.4. CF-EOs Induce ROS-Dependent Apoptosis in Colon Cancer Cells

p53 plays a critical role in regulating cell growth and division. Loss of p53 function is a common event in cancer, and p53-null cells are often more aggressive and treatment-resistant than p53-wt cells. However, natural products that can target cancer cells independently of p53 function are considered a promising approach for cancer therapy. Due to the strong anti-proliferative effect of CF-EOs, we next determined whether they could induce apoptosis. Flow cytometry analysis of apoptosis using Annexin V and PI staining of HCT116 cells showed that CF-EOs dose-dependently induced apoptosis in both cell lines at 50 µg/mL (Figure 4A). Western blotting analyses suggested that CF-EOs increased protein levels of apoptotic markers including Trail, cleaved caspase-3, cleaved caspase-9, and cleaved PARP in both cell lines (Figure 4B). Moreover, we observed an enhanced acetylated-p53 protein in HCT116 p53-wt and an increased acetylated-c-Myc protein in HCT116 p53-null cells, suggesting that CF-EOs provoked apoptosis in a p53-dependent and independent manner (Figure 4B). To investigate whether CF-EOs-mediated apoptosis is also associated with ROS, we pretreated cells with or without NAC before the CF-EOs treatments. Interestingly, protein analysis showed that NAC inhibited the expression of apoptotic markers such as cleaved caspase-3, cleaved caspase-9, and PARP (Figure 4C). Moreover, cells pretreated with NAC significantly attenuated CF-EOs-mediated apoptosis as analyzed by flow cytometry (Figure 4D), suggesting a key role of ROS in CF-EOs-mediated apoptosis. These results highlighted the potential of CF-EOs as a candidate for chemotherapeutic agents regardless of their p53 functionality.

### 3.5. CF-EOs Inhibit the Activity of SIRT1 by Decreasing Its Translation and Inducing Its Autophagic and Proteasomal Degradation

We next explored the molecular events associated with cell death provoked by CF-EOs. The function of SIRT1 in cancer cells is extremely complicated. It can act as a tumor promoter or tumor suppressor, depending on tumor type and its cellular conditions. Therefore, it is important to understand the comprehensive role of SIRT1 in cancer management. We first used data mining software (https://ualcan.path.uab.edu/, and www.kmplot.com, (accessed on 30 October 2023) to examine SIRT1 expression and its relationship with TGCA tumor survival outcomes in colorectal cancer (CRC is referred to here as COAD). We found that high SIRT1 expression at stage (I + II) in 78 male patients with colorectal cancer was associated with poor prognosis for relapse-free survival (RFS) [hazard ratio (HR): 2.11, log-rank *p* = 0.025] (Figure 5A). Therefore, we examined the impact of CF-EOs on SIRT1 activity and protein expression in colon cancer cells. Analysis of the enzymatic activity and protein expression of SIRT1 showed that CF-EOs dose-dependently markedly inhibited SIRT1 activity and protein expression (Figure 5B,C). We further found that pretreatment of HCT116 p53-wt and p53-null cells with the autophagy inhibitor bafilomycin A1 (Baf-A1) and the proteasome inhibitor MG132 prevented the downregulation of SIRT1 protein under treatment with 20 µg/mL and 50 µg/mL of CF-EOs, respectively (Figure 5D,E), suggesting that autophagy and the proteasome pathway may be involved in CF-EO-mediated SIRT1 degradation. These findings indicate that treatment with 20 or 50 µg/mL of CF-EOs leads to the degradation of SIRT1 protein through multiple pathways, including autophagy and proteasomal degradation.

## 4. Discussion

Many standard chemotherapy agents are derived from plants; about 25% of all cancer chemotherapeutics are directly derived from plants and another 25% are chemically modified versions of plant products [42]. For example, the plant product, paclitaxel (brand name Taxol), is approved in the USA as a cancer drug. It targets microtubules and prevents their degradation, which stops cell division and leads to cancer cell death in breast, lung, and ovarian cancer when applied as a monotherapy or in combination with other therapies [43].

Essential oils are complex mixtures of volatile compounds that are extracted from plants and exhibit various biological activities. Currently, almost 3000 essential oils are specified, 300 of which are of commercial importance. They have been used in traditional medicine for centuries and in recent years have attracted the attention of extensive research due to their diverse biological properties [44,45,46]. Some essential oils are being studied for their ability to inhibit tyrosinase and thus have the potential for the development of skin-lightening agents [47]. Essential oils can also induce apoptosis in cancer cells; for example, lime oil from *Citrus aurantifolia* contains 22 volatile compounds and reportedly induces apoptosis in colon adenocarcinoma (SW-480 cells) through DNA fragmentation and caspase-3 activation [21]. Essential oil compounds were shown to induce cleavage of poly (ADP-ribose) polymerase-1 (PARP) in HCT116 p53-wt and p53-null cells, which is a hallmark of apoptosis commonly induced by anticancer drugs [48,49]. The EOs component, thymol, reportedly exerts anti-proliferative activity, increases the generation of ROS, and induces apoptosis by activating PARP and caspases-3 and -9 in acute promyelotic leukemia cells [50]. In our study, we showed that the pretreatment of NAC significantly reduced CF-EO-mediated autophagy and apoptosis, suggesting oxidative stress is involved in the system. Oxidative stress/damage is often associated with ROS. ROS include not only the oxygen radicals and non-radical oxidizing agents, but also peroxynitrite (ONOO−/ONOOH) and nitrogen dioxide radical (NO2^•^) [51]. Although, at this point, we are still vague on what types of ROS act as the ultimate signal for the anticancer properties of CF-EOs. With the guidelines reported by Murphy et al., it could be achievable to assess oxidative events to further understand the biological importance of CF-EOs.

Here, we show for the first time that CF-EOs significantly induce anti-proliferative activity, autophagy at 20, 50 µg/mL and apoptosis at 50 µg/mL in a CRC cell line regardless of p53, as analyzed by MTT, real-time cell monitoring, and flow cytometry. Western blot analysis also supported these findings, showing that CF-EOs increased the level of ULK1, cleavages of LC3-II at 20, 50 µg/mL, caspase-3, and caspase-9, and the activation of PARP at 50 µg/mL in both cell lines. A cytotoxic autophagy can directly or indirectly trigger cancer cell death in response to anticancer drugs [52]. Nuclear localization of the autophagy initiation kinase Unc-51-like kinase (ULK1)/ATG1 induces autophagic cell death [53]. Protein acetylation plays an important role in autophagy [54]. Moreover, we have previously shown that the natural compound capsaicin significantly induced acetylation of ULK1 and triggered autophagy in p53-mutated HSC-3 oral cancer cells [55].

Using chemotherapeutic drugs as cancer treatments causes adverse effects in patients and those patients often develop drug resistance. To find an alternative to chemotherapy that can improve the survival of cancer patients and lower those side effects, researchers have focused on the employment of natural resources. Nowadays, natural substances are used as aids to reduce the side effects. For example, to reduce the dosage of doxorubicin, *Zataria multiflora* essential oil (ZEO) was used simultaneously, which sensitized the prostate cancer cells to endure ROS formation and apoptosis [56]. Falih et al. demonstrated that the combination of eucalyptus oil and retinoic acid, as found in leafy vegetables, carrots, and oranges, has a synergistic effect leading to the induction of external and internal apoptosis in SK-GT-4 cells [57]. In the HCT116 xenograft model, mice treated with the EO of turmeric (ETO-Cur) and the tocotrienol-rich fraction (TRF) of vitamin E isomers showed synergistic inhibition of tumor volume [3]. Moreover, studies have shown that doxorubicin in combination with *Pituranthos chloranthus* (PC) EO or *Teucrium ramosissimum Desf.* (TR) induces a synergistic effect in MES-SA/Dx5 cells and reduces the resistance index [58]. Al-Otaibi et al. showed that the combination of *Teucrium polium L.* essential oil nanoemulsion and oxaliplatin had a synergistic effect with different p53 status colon cancer cells and induced a greater percentage of apoptosis [59]. In addition, combination with dabrafenib and/or trametinib *Melaleuca alternifolia* (tea tree oil) synergistically reduced cell viability and prompted apoptosis in melanoma [60]. However, in this present study, we did not compare the anticancer activity of CF-EOs with a reference compound. It would also be interesting to examine whether the combination of CF-EOs with known therapeutic agents exhibit synergistic effects.

SIRT1 belongs to the sirtuin lineage and functions as an NAD^+^-dependent deacetylase. It deacetylates histone and non-histone proteins, which ultimately controls their activities and functions. As a result, it influences various cellular processes such as apoptosis and autophagy. For example, inhibition of SIRT1 enhances the acetylation of p53 and c-Myc, thereby activating their tumor suppressor activities and inducing cancer cell death. The forthcoming index shows that overexpression of SIRT1 is associated with poor prognosis in many types of solid tumors, including breast, prostate, bladder, and colorectal carcinoma [61]. Therefore, targeting SIRT1 with anticancer drugs or other therapies is a possible strategy to eliminate cancer cells [62,63,64,65]. Pharmacological inhibition or genetic knockdown of SIRT1 significantly reduced the colony-forming ability of colon cancer cells [61]. In our previous works, we showed that SIRT1 deacetylase is crucial for the capability of natural products, such as capsaicin and heliomycin, to induce cell death [36,65]. Here, we found that CF-EOs significantly enhanced SIRT1 downregulation in both HCT116 p53-wt and p53-null cells, whereas the inhibitors of the pathways markedly reversed CF-EO-mediated SIRT1 downregulation.

## 5. Conclusions

Our findings collectively demonstrate that CF-EOs elevated ROS production and triggered ROS-dependent autophagy and apoptosis in both p53-wt and null cell lines and thereby inhibited the growth of cancer cells. We further provide evidence to support the idea that CF-EO-targeted SIRT1 inhibition may be a promising strategy in colon cancer management.

## Figures and Tables

**Figure 1 antioxidants-13-00284-f001:**
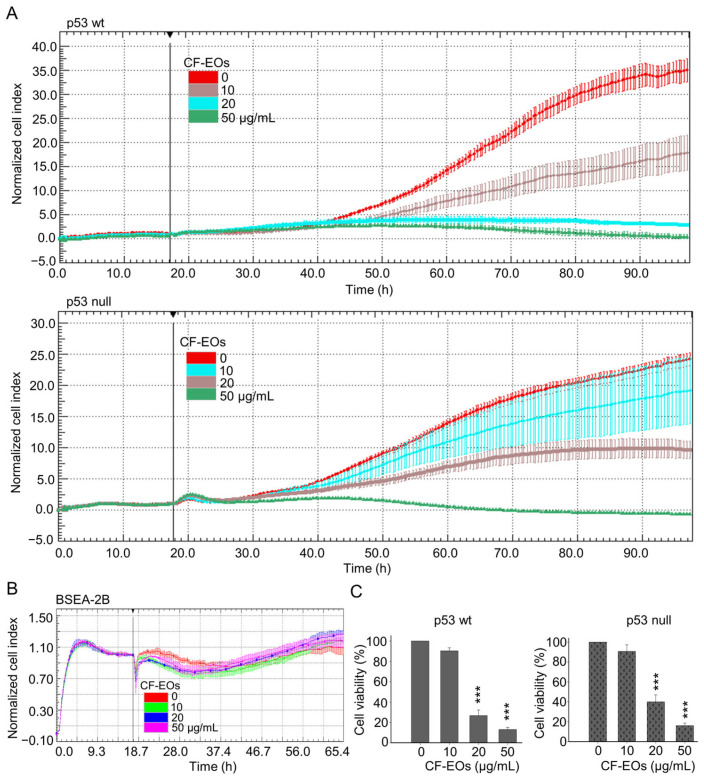
CF-EOs repress the growth of colon cancer cells but not of non-tumorigenic cells. HCT116 p53-wt, HCT116 p53-null, and BSEA-2B cells were treated with different concentrations of CF-EOs for different durations. (**A**,**B**) Cell growth was observed using an impedance measurement technique. Cells were treated with various concentrations of CF-EOs after overnight growth. Represented graphs are normalized cell index values measured. A significant difference in p53-wt and-null cell lines was calculated using a student’s *t*-test followed by a two-tailed test in experimental groups vs control. (**C**) Cell viability was determined by MTT assay. A significant difference was calculated in experimental groups vs control (*** *p* < 0.001).

**Figure 2 antioxidants-13-00284-f002:**
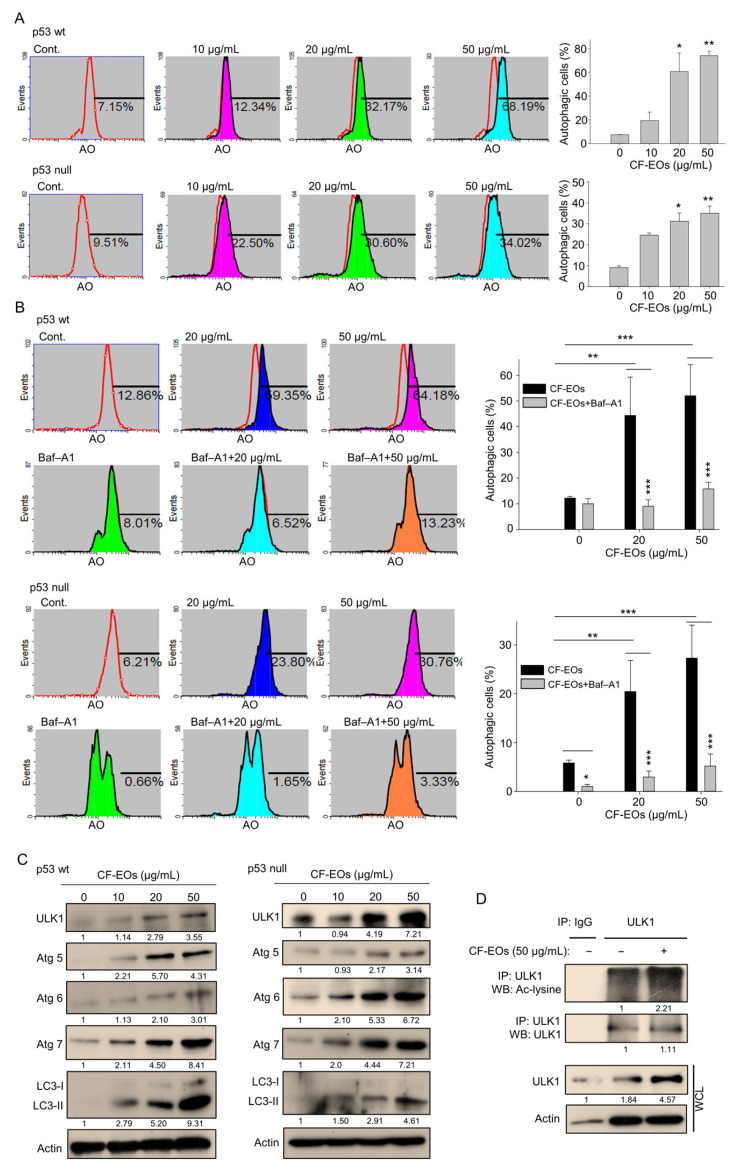
CF-EOs effectively induce autophagy in HCT116 p53-wt and p53-null cells. (**A**) Acridine orange (AO) staining of HCT116 p53-wt and HCT116 p53-null cells treated with different concentrations of CF-EOs for 18 h. AO staining enables visualization of acidic vesicles, which are a hallmark of autophagy; staining was analyzed by flow cytometry and the results are expressed as a percentage of autophagic cells from at least three independent experiments (* *p* < 0.05, ** *p* < 0.01). (**B**) AO staining of HCT116 p53-wt and HCT116 p53-null cells pretreated with or without 10 nM Baf-A1 for 1 h and then treated with 20 or 50 µg/mL CF-EOs for 18 h. Cells were analyzed by flow cytometry and the results are expressed as a percentage of autophagic cells. A significant difference was calculated in experimental groups vs control (* *p* < 0.05, ** *p* < 0.01, *** *p* < 0.001). (**C**) HCT116 p53-wt and HCT116 p53-null cells were treated with DMSO and different concentrations of CF-EOs for 18 h. Protein expression was analyzed by Western blotting. β-Actin was used as an internal loading control. (**D**) Cells were incubated with 1 mM nicotinamide (NIC) for 1 hr prior to harvest. Lysates from HCT116 p53-wt cells were immunoprecipitated with nonimmune IgG or an antibody against ULK1, and the bound proteins were detected by Western blotting with anti-acetylated lysine or anti-ULK1 antibodies.

**Figure 3 antioxidants-13-00284-f003:**
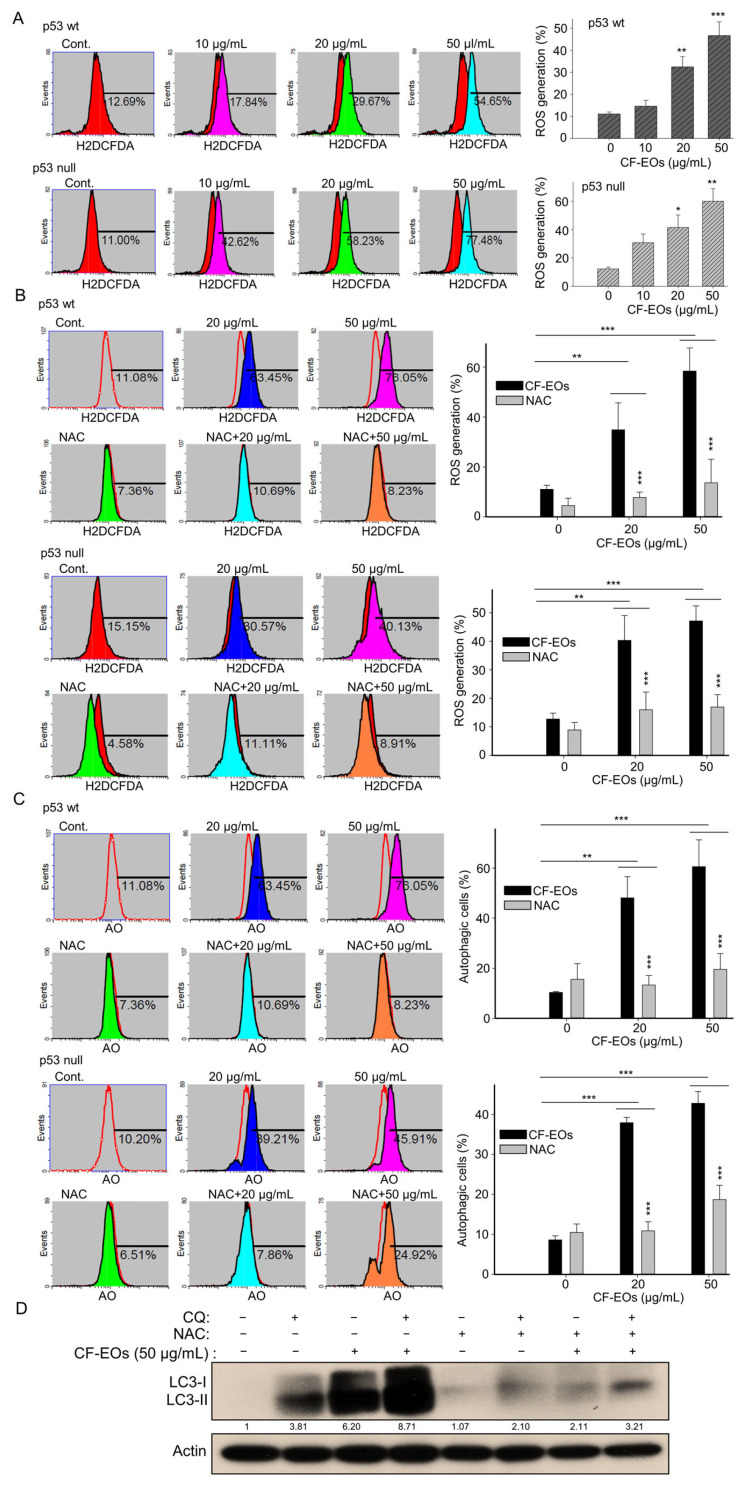
CF-EOs-induced autophagy of HCT116 p53-wt and p53-null cells is ROS-dependent. (**A**) HCT116 p53-wt and HCT116 p53-null cells were treated with different concentrations of CF-EOs for 1 h. Cells were stained with ROS dye for 30 min at 37 °C. The ROS formation was determined by flow cytometry and expressed as a percentage of ROS generation. Representative images and values (means ± SDs) are from no less than three independent experiments (* *p* < 0.05, ** *p* < 0.01, *** *p* < 0.001). (**B**) Cells were pretreated with or without 10 mM NAC for 1 h before being exposed to 20 or 50 µg/mL CF-EO for 1 h. Representative images and values (means ± SDs) are from no less than three independent experiments (** *p* < 0.01, *** *p* < 0.001). (**C**) AO staining of HCT116 p53-wt and HCT116 p53-null cells pretreated with or without 10 nM NAC for 1 h and then treated with 20 or 50 µg/mL CF-EO for 18 h. Cells were analyzed by flow cytometry and the results are expressed as a percentage of autophagic cells. A significant difference was calculated in experimental groups vs control (** *p* < 0.01, *** *p* < 0.001). (**D**) HCT116 p53-wt cells were co-exposed in the presence or absence of 50 µg/mL CF-EOs, NAC, and the autophagosome-lysosome binding inhibitor CQ (25 μM) for 18 h. Aliquots of cell lysates were resolved by SDS-PAGE and analyzed for protein expression by Western blotting. β-actin was used as an internal loading control to monitor for equal loading.

**Figure 4 antioxidants-13-00284-f004:**
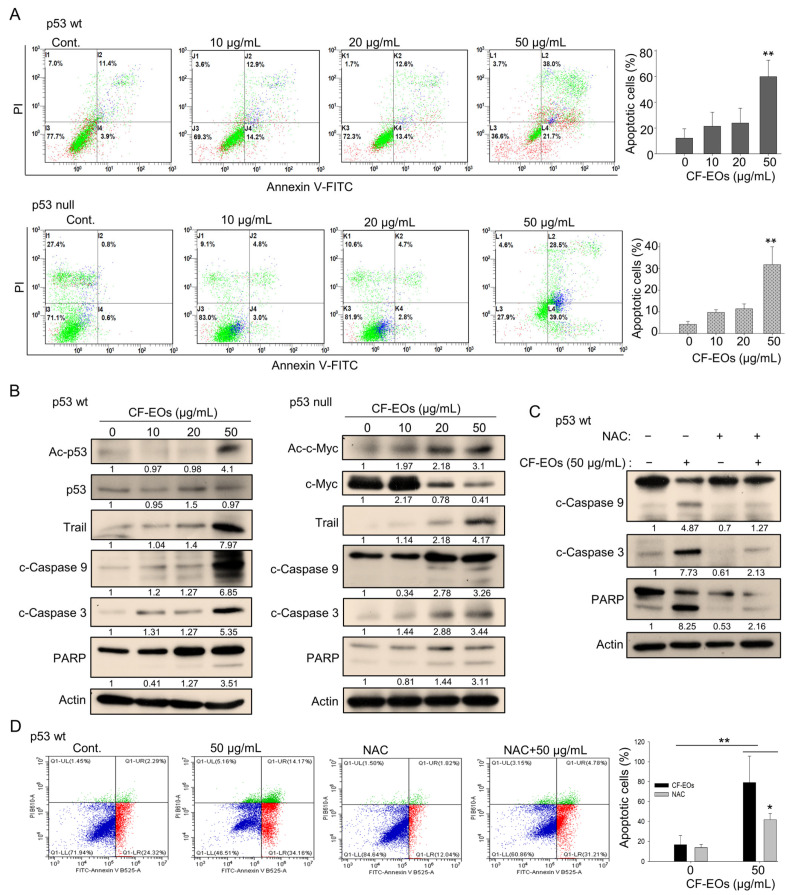
CF-EOs trigger ROS-dependent apoptosis in colon cancer cells. (**A**) Flow cytometric analysis of apoptosis in HCT116 p53-wt and p53-null cells treated with different concentrations of CF-EOs for 24 h. Annexin V and propidium iodide (PI) were used to stain the cells. Cells were analyzed by flow cytometry and the percentage of apoptotic cells (Annexin V-positive and PI-negative cells) is shown. Representative images are provided from at least three independent experiments. A significant difference was calculated in experimental groups vs control (** *p* < 0.01). (**B**) Western blot analysis of apoptosis-related proteins in HCT116 p53-wt and p53-null cells treated with DMSO (vehicle control) and various concentrations of CF-EOs for 24 h. Protein levels of Ac-p53, p53, Ac-c-Myc, c-Myc, Trail, cleaved caspase-3, cleaved caspase-9, and cleaved PARP were analyzed. β-actin was used as an internal loading control. (**C**,**D**) HCT116 p53-wt cells were pretreated with or without 10 nM NAC for 1 h and then treated with 50 µg/mL CF-EO for 24 h. (**C**) Protein levels of cleaved caspase-3, cleaved caspase-9, and cleaved PARP were analyzed by western blot. β-actin was used as an internal loading control. (**D**) Cells were analyzed by flow cytometry and the results are expressed as a percentage of apoptotic cells. A significant difference was calculated in experimental groups vs control (* *p* < 0.05, ** *p* < 0.01).

**Figure 5 antioxidants-13-00284-f005:**
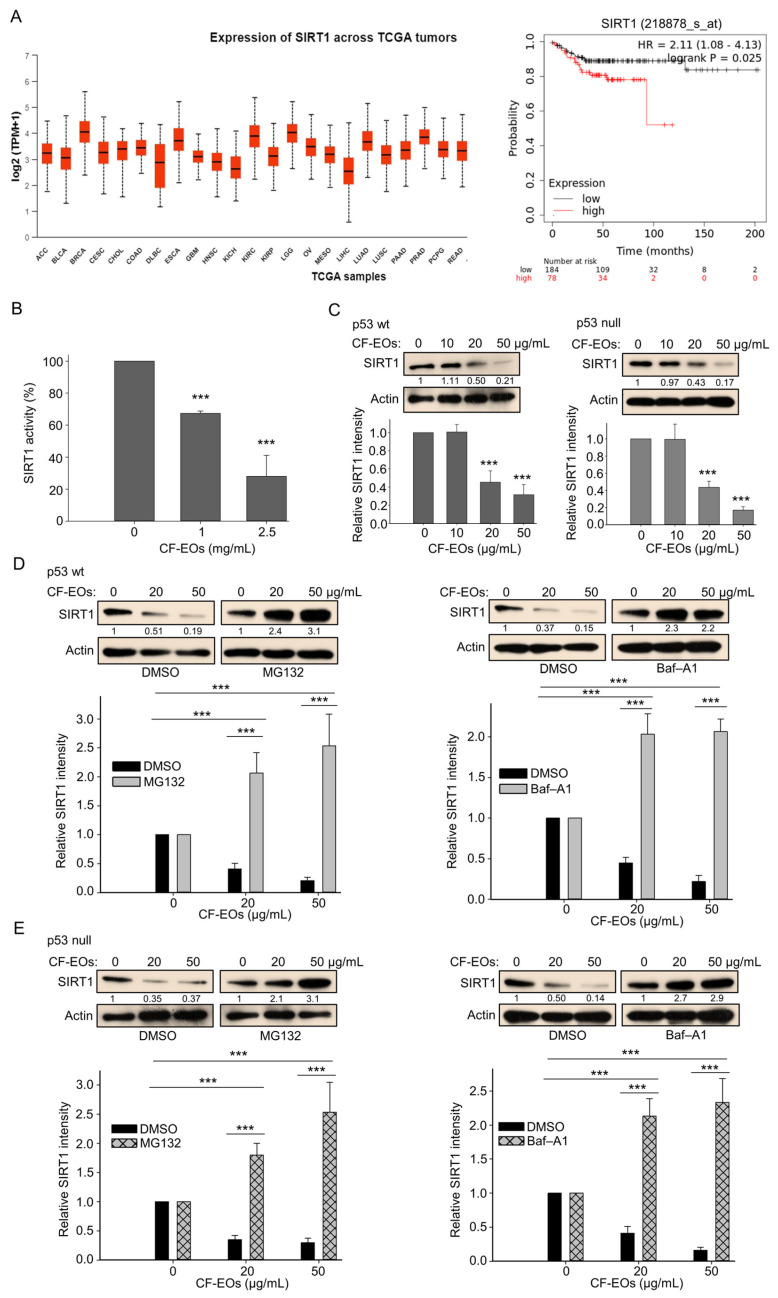
CF-EOs inhibit SIRT1 activity and protein expression in colon cancer cells. (**A**) Expression of SIRT1 and survival diagrams (https://ualcan.path.uab.edu/ and www.kmplot.com (accessed on 30 October 2023) of the relationship between SIRT1 expression and relapse-free survival in 262 patients with stage (I + II) CRC. (**B**) CF-EOs attenuate cellular SIRT1 activity, as analyzed using a SIRT1 Activity Assay Kit (Fluorometric) with control or CF-EOs-exposed recombinant SIRT1. Representative images and values (means ± SDs) are from at least three independent experiments (*** *p* < 0.001). (**C**) Western blot analysis of SIRT1 protein expression in HCT116 p53-wt and HCT116 p53-null cells treated with different concentrations of CF-EOs for 24 h. β-Actin was used as an internal loading control. Quantification of SIRT1 levels is provided from at least three independent experiments. Values (means ± SDs) are from at least three independent experiments (*** *p* < 0.001). (**D**,**E**) SIRT1 protein expression in HCT116 p53-wt and HCT116 p53-null cells pretreated with or without 10 nM Baf-A1 or 5 μM MG132 for 1 h followed by treatment with 20 or 50 µg/mL CF-EOs for 18 or 24 h, as analyzed by Western blotting. β-Actin was used as an internal loading control. Representative images are provided from at least three independent experiments. Values (means ± SDs) are from at least three independent experiments (*** *p* < 0.001).

**Table 1 antioxidants-13-00284-t001:** Composition of wood essential oils from *Calocedrus formosana*.

RT (min) ^a^	Constituent	Concentration (%)	KI ^b^	Identification ^c^
8.83	Camphene	0.78	949	MS, KI
10.12	α-Methylstyrol	0.13	980	MS, KI
11.29	3-Carene	0.19	1007	MS, KI
11.56	1,4-Cineol	0.22	1014	MS, KI, ST
11.64	α-Terpinene	0.2	1016	MS, KI
11.75	*o*-Cymene	0.1	1019	MS, KI
11.96	β-Cymene	2.12	1024	MS, KI
12.18	Limonene	1.13	1029	MS, KI, ST
13.45	γ-Terpinene	0.51	1058	MS, KI
14.47	p-Cymenene	0.8	1080	MS, KI
14.67	α-Terpinolene	1.01	1084	MS, KI
14.87	*p*-Cymenene	0.86	1088	MS, KI
17.42	Camphor	0.36	1145	MS, KI, ST
19	4-Terpineol	3.2	1178	MS, KI, ST
19.38	α-Terpineol	1.47	1185	MS, KI, ST
19.68	L-α-Terpineol	5.92	1191	MS, KI, ST
20.17	Verbenone	1.09	1201	MS, KI, ST
20.33	(–)-cis-Verbenone	0.23	1204	MS, KI
21.82	Anisole	0.24	1232	MS, KI
25.47	Thymol	0.61	1294	MS, KI, ST
27.88	Shonanic acid	34.86	1329	ST
28.4	Chaminic acid	10.06	1336	ST
29.83	Myrtenoic acid	3.84	1355	MS, KI
31.35	Thujic acid	10.2	1374	ST
34.2	α-Bourbonene	4.02	1408	MS, KI
41.04	Germacrene D	0.31	1490	MS, KI
42.4	cis-γ-Cadinene	0.88	1505	MS, KI

^a^ retention time. ^b^ Kovats index on a DB-5MS column in reference to n-alkanes. ^c^ MS, NIST, and Wiley library spectra and the literature; KI: Kovats index; ST: authentic standard compound.

## Data Availability

Data is contained within the article.

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
