# Peer review of "Calocedrus formosana Essential Oils Induce ROS-Mediated Autophagy and Apoptosis by Targeting SIRT1 in Colon Cancer Cells"

_antioxidants, 2024, doi:10.3390/antiox13030284_

Round 1

Reviewer 1 Report

Comments and Suggestions for Authors

In their paper titled “Calocedrus formosana essential oils induce ROS-mediated autophagy and apoptosis by targeting SIRT1 in colon cancer cells” Islam et al. evaluated the effects of essential oils extracted from Calocedrus formosana (CF-EOs) in p53- wild-type and p53- null colon cancer cells. The authors provided evidence that CF-EOs promoted an increase in ROS level inducing  autophagy and apoptosis in both p53-wild-type and -null cell lines. Such an effect could be ascribed to the inhibition of SIRT1 activity.

However,  to consider this paper suitable for publication, the authors should address the following points:

 ·       Please, provide evidence that autophagy and apoptotic cell death are correlated to ROS generation. The authors could ascertain whether the administration of NAC or glutathione can prevent the autophagic flux and apoptosis.

·         At page 6: lanes 232-233 authors report that CF-EOs significantly decreased the cell growth. The p value should be reported in the sentence. In the lane 255 edit p53-depednent in p53-dependent.

·         In figure 2A, the effect of 10 mg/ml CF-EO on in p53- wild-type colon cancer cells for AnnV/PI double staining analyses is reported twice, while 20 mg/ml CF-EO is missing. Please check data in this figure.

·         In the paper, the authors demonstrated that CF-EOs were able to induce autophagy in both p53- wild-type and p53- null colon cancer cells.  The role of p53 should be better clarified and discussed in the text by authors, since although p53 can clearly trigger apoptosis, its seems to play a controversial role in autophagy (for details see: Notaro et al. Antioxidants 2023, 12,1292; Scherz-Shouval, R. et al. Proc. Natl. Acad. Sci. USA 2010, 107, 18511).

·        The description of SIRT1 activity in tumor models should be moved in the discussion section and authors should discuss the meanining of the downregulation in SIRT1 activity in this section of the paper.

·        The effects of the major compounds found in CF-EOs should be tested alone and in combination on p53- wild-type and p53- null colon cancer cells to ascertain which phytochemicals are responsible for the observed mechanism.

Comments on the Quality of English Language

Since some typing errors are present in the text, minor editing of English language is required.

Author Response

Reviewer 1# In their paper titled “Calocedrus formosana essential oils induce ROS-mediated autophagy and apoptosis by targeting SIRT1 in colon cancer cells” Islam et al. evaluated the effects of essential oils extracted from Calocedrus formosana (CF-EOs) in p53- wild-type and p53- null colon cancer cells. The authors provided evidence that CF-EOs promoted an increase in ROS level inducing autophagy and apoptosis in both p53-wild-type and -null cell lines. Such an effect could be ascribed to the inhibition of SIRT1 activity. However, to consider this paper suitable for publication, the authors should address the following points:

  1. Please, provide evidence that autophagy and apoptotic cell death are correlated to ROS generation. The authors could ascertain whether the administration of NAC or glutathione can prevent the autophagic flux and apoptosis.

Author’s response: Thank you very much for your valuable time and suggestion. In response to your suggestions, we have performed new experiments and revised our manuscript. In page 13, line 379-386, we have added NAC administration apoptosis results. “To investigate whether CF-EOs-mediated apoptosis is also associated with ROS, we pretreated cells with or without NAC before the CF-EO treatments. Interestingly, protein analysis showed that NAC inhibited the expression of apoptotic markers such as cleaved caspase-3, cleaved caspase-9, and PARP (Figure 4C). Moreover, cells pretreated with NAC significantly attenuated CF-EOs-mediated apoptosis as analyzed by flow cytometry (Figure 4D), suggesting a key role of ROS in CF-EO-mediated apoptosis. These results highlighted the potential of CF-EOs as a candidate for chemotherapeutic agents regardless of their p53 functionality.”

In page 11, line 338-347, “we have added NAC administration autophagy flux results. “To investigate the role of ROS in autophagy flux, HCT116 p53-wt cells were cotreated with or without autophagosome lysosome binding inhibitor chloroquine (CQ), NAC, and 50 µg/ml CF-EOs. Concomitant treatment with CQ and 50 µg/ml CF-EOs increased LC3-II protein levels to a greater extent than in the CF-EOs alone group, indicating increased accumulation of autophagosomes and enhancement of autophagic flux. Surprisingly, concomitant treatment with the ROS inhibitor NAC markdely attenuated LC3-II expression induced by 50 µg/ml CF-EOs alone as well as by the combination of CQ and 50 µg/ml CF-EOs, suggesting that ROS is essential for autophagosomes and autophagy flux as analyzed by Western blotting (Figure 3D).”

  1. At page 6: lanes 232-233 authors report that CF-EOs significantly decreased the cell growth. The p value should be reported in the sentence. In the lane 255 edit p53-depednent in p53-dependent.

Author’s response: Thank you very much for your valuable time and suggestion. In response to your suggestions, we have added the p-value (***P<0.000) in page 8, line 287 in our current version of manuscript. The original p value as follows:

The p value for 10 µg/ml is 3.23x10-07, for 20 µg/ml is 1.38x10-08, for 50 µg/ml is 4.37x10-09 in p53 wt HCT116 cells. The p value for 10 µg/ml is 0.0001, for 20 µg/ml is 3.85x10-05, for 50 µg/ml is 3.65x10-05 in p53 null HCT116 cells.

In the lane 255 edit p53-depednent: We are sincerely sorry for our typo and the correction has been made in page 13 line 379. p53-depednent

  1. In figure 2A, the effect of 10 mg/ml CF-EO on in p53- wild-type colon cancer cells for AnnV/PI double staining analyses is reported twice, while 20 mg/ml CF-EO is missing. Please check data in this figure.

Author’s response: Thank you very much for your valuable time. We are sincerely sorry and apologize for our careless mistake. The correction has been made in figure 4A (which was figure 2A in previous draft). 

  1. In the paper, the authors demonstrated that CF-EOs were able to induce autophagy in both p53- wild-type and p53- null colon cancer cells.  The role of p53 should be better clarified and discussed in the text by authors, since although p53 can clearly trigger apoptosis, its seems to play a controversial role in autophagy (for details see: Notaro et al. Antioxidants 2023, 12,1292; Scherz-Shouval, R. et al. Proc. Natl. Acad. Sci. USA 2010, 107, 18511).

Author’s response:  Thank you very much for your time and valuable suggestion. In response to your suggestions, we have added the role of p53 in apoptosis and autophagy. In page 3, line 99-102 “p53 is known as the "guardian of the genome", acting as a tumor suppressor. Studies have shown that p53-mediated apoptosis is important in the prevention of tumor development [38]. However, the role of p53 in autophagy is more complex and paradoxical, depending on its subcellular location [39-41].

In page 17, line 470-475. “A cytotoxic form of autophagy can directly or indirectly promote cancer cell death in response to anticancer drugs [51]. Nuclear localization of the autophagy initiation kinase Unc-51-like kinase (ULK1)/ATG1 induces autophagic cell death [52]. Protein acetylation plays an important role in autophagy [53]. In our previous study, we have shown that the natural compound capsaicin significantly induces acetylation of ULK1 in p53-mutated HSC-3 oral cancer cells and triggers autophagy [54].”

In addition, to confirm the role of p53 in autophagy in our system, we performed immunoprecipitation assay (page10, line 312-314). “To further confirm the autophagic pathway induced by CF-EOs, we performed an immunoprecipitation assay and validated an enhancement of acetylated ULK1 by CF-EOs (Figure 2D).”

  1. The description of SIRT1 activity in tumor models should be moved in the discussion section and authors should discuss the meanining of the downregulation in SIRT1 activity in this section of the paper.

Author’s response: Thank you very much for your time and valuable suggestion. In response to your suggestions, we relocated SIRT1 paragraph in the discussion section (page 18, line 495-508). “SIRT1 belongs to the sirtuin lineage and functions as an NAD+-dependent deacetylase. It deacetylates histone and non-histone proteins, which ultimately controls their activities and functions. As a result, it influences various cellular processes such as apoptosis and autophagy. For example, inhibition of SIRT1 enhances the acetylation of p53 and c-Myc, thereby activating their tumor-suppressor activities and inducing cancer cell death. The forthcoming index shows that overexpression of SIRT1 is associated with poor prognosis in many types of solid tumors, including breast, prostate, bladder, and colorectal carcinoma [60]. Therefore, targeting SIRT1 with anticancer drugs or other therapies is therefore a possible strategy to destroy cancer cells [61-64]. Pharmacological inhibition or genetic knockdown of SIRT1 significantly reduced the colony-forming ability of colon cancer cells [60]. In previous studies, we showed that SIRT1 deacetylase is crucial for the abilities of natural products, such as capsaicin, and heliomycin, to induce cell death [36, 64]. In our current study, we found that CF-EOs significantly induced the downregulation of SIRT1 in both HCT116 p53-wt and p53-null cells, whereas proteasomal degradation and autophagy inhibitor significantly reversed CF-EOs-induced SIRT1 downregulation.”

  1. The effects of the major compounds found in CF-EOs should be tested alone and in combination on p53- wild-type and p53- null colon cancer cells to ascertain which phytochemicals are responsible for the observed mechanism.

Author’s response: Thank you very much for your time and suggestions. In our study we did not use the individual purified compound but focus on using the mixture of CF-EOs. We intended to exploit the combination of all components in CF-EOs, rather than individual purified compound in cancer management. There are precedents of utilizing such mixtures to achieve synergistic effect on reducing cancer. Those examples are now incorporated into our revised Discussion section (page 17).  Using chemotherapeutic drugs as cancer treatments causes adverse effects in patients and those patients often develop drug resistance. To find an alternative to chemotherapy that can improve the survival of cancer patients, researchers have focused on natural resources. Nowadays, natural substances are used as aids to reduce the side effects. For example, to reduce the dosage of doxorubicin, Zataria multiflora essential oil (ZEO) was used simultaneously, which sensitized the prostate cancer cells to endure ROS formation and apoptosis [55]. Falih et al., demonstrated that the combination of eucalyptus oil and retinoic acid, as found in leafy vegetables, carrots and oranges, has a synergistic effect leading to the induction of external and internal apoptosis in SK-GT-4 cells [56]. In the HCT116 xenograft model, mice treated with the EO of turmeric (ETO-Cur) and the tocotrienol-rich fraction (TRF) of vitamin E isomers showed synergistic inhibition of tumor volume [3]. Moreover, studies have shown that doxorubicin in combination with Pituranthos chloranthus (PC) EO or Teucrium ramosissimum Desf. (TR) induces a synergistic effect in MES-SA/Dx5 cells and reduces the resistance index [57]. Al-Otaibi et. al, shown that the combination of Teucrium polium L. essential oil nanoemulsion and oxaliplatin had a synergistic effect with different p53 status colon cancer cells and induced a greater percentage of apoptosis [58]. In addition, combination with dabrafenib and/or trametinib Melaleuca alternifolia (tea tree oil) synergistically reduced cell viability and prompted by apoptosis in melanoma [59].”

Comments on the Quality of English Language

Since some typing errors are present in the text, minor editing of English language is required.

Author’s response: Thank you very much for your time and valuable suggestions. We are sincerely sorry for our careless mistakes. In response to your suggestions, we have revised our manuscript thoroughly and the correction has been made.

Reviewer 2 Report

Comments and Suggestions for Authors

The manuscript investigates the effects of Calocedrus formosana essential oils on two colon cancer cell lines. After reading this work, I have some observations:

1.      I recommend rereading the text for typos and grammatical mistakes.

2.      Add the aim of the study to the abstract.

3.      Findings shouldn’t be part of the introduction. Relocate lines 96-99 to the results and discussion or the conclusions.

4.      Add the herbarium voucher number.

5.      What were the plant material-to-water ratio and the distillation rate?

6.      The GCMS analysis of the extracted oil is missing from the methods. Add the details of the solvent used, instrument parameters, and compound identification.

7.      Lines 214-218: Add details of the corresponding methods used to the methods section.

8.      Avoid using references in the Results section. Rewrite these parts.

9.      All Figures, Schemes, and Tables should be inserted into the main text close to their first citation.

10.   The discussion part is not strong enough. Consolidate the arguments with a focus on the findings of this study.

Comments on the Quality of English Language

The work would benefit from rereading for typos and grammatical mistakes.

Author Response

Reviewer#2

The manuscript investigates the effects of Calocedrus formosana essential oils on two colon cancer cell lines. After reading this work, I have some observations:

  1. I recommend rereading the text for typos and grammatical mistakes.

Author’s response: Thank you very much for your time and valuable suggestions. We are sincerely sorry for our careless mistakes. In response to your suggestions, we have revised our manuscript thoroughly and the correction has been made.

  1. Add the aim of the study to the abstract.

Author’s response: Thank you very much for your time and valuable suggestions. We are sincerely sorry for our careless mistakes. In response to your suggestions, we have added the aim in abstract (page 1 line 19-21). “However, the anti-cancer effects of CF-EOs are not yet fully understood. Therefore, the aim of the present study was to investigate the potential anticancer mechanism underlying CF-EOs in colon cancer cells.”

  1. Findings shouldn’t be part of the introduction. Relocate lines 96-99 to the results and discussion or the conclusions.

Author’s response: Thank you very much for your time and valuable suggestions. In response to your suggestions, we have relocated the findings part in discussion (page 19, line 533-536). “Here, we also found that CF-EOs elevated ROS production to induce autophagy and apoptosis in both p53-wild-type (wt) and -null cell lines and thereby inhibit the growth of cancer cells. Our findings collectively suggest that CF-EOs trigger ROS-dependent autophagy and apoptosis in CRC, and thus may have anticancer potential in this disease.”

  1. Add the herbarium voucher number.

Author’s response: Thank you very much for your time and valuable suggestions. In response to your suggestion, we have added the voucher number (Page 3, line 130-131). “The voucher specimen (Y.S. Tseng 4325, TCF) was placed in the herbarium of the same university.”

  1. What were the plant material-to-water ratio and the distillation rate?

Author’s response:  Thank you very much for your time and valuable suggestions. In response to your suggestions, we have included the plant material-to-water ratio and the distillation rate (Page 3, line 131-134). “400 g of air-dried wood crisps and 1.6 L (ratio 1:4) water were hydro-distilled in a Clevenger apparatus for 6 hours and the oil content was then determined. The average distillation rate of essential oils was approximately 1.34 g per hour.

  1. The GCMS analysis of the extracted oil is missing from the methods. Add the details of the solvent used, instrument parameters, and compound identification.

Author’s response: Thank you very much for your time and valuable suggestions. We are sincerely sorry for our careless mistake. In response to your suggestions, we have included the GC-MS analysis in the methods section (page 4, line 137-149). “GC-MS was used to analyze the chemical composition of the CF-EOs using an ITQ 900 mass spectrometer in conjunction with a DB-5MS column. The temperature of the GC oven was initially set at 40 ⁰C for 3 minutes, then increased to 180 ⁰C at a rate of 3 ⁰C/min and then increased to 20 ⁰C /min until 280 ⁰C was reached and held for 5 minutes. The injection temperature was maintained at 240 ⁰C, while the ion source temperature remained at 200 ⁰C during the analysis. Helium was used as the carrier gas at a rate of 1 ml/min, and the mass scanning range was 40-600 m/z. Standard databases such as Wiley/NBS Registry of mass spectral databases (V. 8.0, Hoboken, NJ, USA) and the National Institute of Standards and Technology (NIST) Ver. 2.0 GC/MS libraries were used to identify the volatile compounds. The Kovats index (KI) was calculated from the retention times of a compound using a homologous series of n-alkanes C9-C24. Standards were co-injected to identify the main components.”

  1. Lines 214-218: Add details of the corresponding methods used to the methods section.

Author’s response: Thank you very much for your time and valuable suggestions. We are sincerely sorry for our careless mistake. In response to your suggestions, we have included the “identification of unknown compounds” in the methods section (page 4, line 150-167). “The three unique CF-EOs compounds such as shonanic acid, thujic acid, and chaminic acid were purified and identified by chromatography and spectroscopy techniques as described previously [19]. In brief, Using the Agilent 1100 HPLC system with UV detector, the CF-EOs were further separated. For the establishment of COSMOSIL C18-AR-II (10 mm I.D. × 250 mm, Nacalai Tesque Inc., San Diego, CA, USA), the solvent system MeOH/H2O was used. The gradient elution profile at a flow rate of 1.6 ml/min was as follows: 0 to 3 min, MeOH:H2O = 55:45; 3 to 13 min, MeOH:H2O = 75:25; 13 to 18 min, MeOH:H2O = 75:25; 13 to 18 min, MeOH:H2O = 75:25; 18 to 25 min, MeOH:H2O = 85:15; 25 to 30 min, MeOH:H2O = 85:15; 30 to 33 min, MeOH:H2O = 100:0; 5; 33 to 40 min, MeOH:H2O = 100:0. The wavelength of the UV detector was set to 254 nm. To identify the structure, the compounds were dissolved in deuterated chloroform (d-chloroform; CDCl3). 1H, 13C and 2D NMR spectra were recorded on a Bruker AVANCE III NMR spectrometer (Bruker, Billerica, MA, USA), with 1H data acquired at 400 MHz and 13C data acquired at 100 MHz using standard experiments from the Bruker pulse program library. High-resolution mass spectrometry (HRMS) was determined using an LTQ Orbitrap XL (Thermo Fisher Scientific, Waltham, MA, USA). The detected compounds were evaluated by MS and 400 MHz NMR.”

  1. Avoid using references in the Results section. Rewrite these parts.

Author’s response: Thank you very much for your time and valuable suggestions. In response to your suggestions, we have removed the references from the Results section and the correction has been made.

  1. All Figures, Schemes, and Tables should be inserted into the main text close to their first citation.

Author’s response: Thank you very much for your time and valuable suggestions. In response to your suggestions, we have inserted the table and all figures into the main text close to the first citation.

  1. The discussion part is not strong enough. Consolidate the arguments with a focus on the findings of this study.

Author’s response: Thank you very much for your time and valuable suggestions. In response to your suggestions, we have included the following paragraphs in the discussion section in our current version of the manuscript.

In page 17, line 470-475. “A cytotoxic form of autophagy can directly or indirectly promote cancer cell death in response to anticancer drugs [51]. Nuclear localization of the autophagy initiation kinase Unc-51-like kinase (ULK1)/ATG1 induces autophagic cell death [52]. Protein acetylation plays an important role in autophagy [53]. In our previous study, we have shown that the natural compound capsaicin significantly induces acetylation of ULK1 in p53-mutated HSC-3 oral cancer cells and triggers autophagy [54].”

In page 17-18, line 476-494. “Using chemotherapeutic drugs as cancer treatments causes adverse effects in patients and those patients often develop drug resistance. To find an alternative to chemotherapy that can improve the survival of cancer patients, researchers have focused on natural resources. Nowadays, natural substances are used as aids to reduce the side effects. For example, to reduce the dosage of doxorubicin, Zataria multiflora essential oil (ZEO) was used simultaneously, which sensitized the prostate cancer cells to endure ROS formation and apoptosis [55]. Falih et al., demonstrated that the combination of eucalyptus oil and retinoic acid, as found in leafy vegetables, carrots and oranges, has a synergistic effect leading to the induction of external and internal apoptosis in SK-GT-4 cells [56]. In the HCT116 xenograft model, mice treated with the EO of turmeric (ETO-Cur) and the tocotrienol-rich fraction (TRF) of vitamin E isomers showed synergistic inhibition of tumor volume [3]. Moreover, studies have shown that doxorubicin in combination with Pituranthos chloranthus (PC) EO or Teucrium ramosissimum Desf. (TR) induces a synergistic effect in MES-SA/Dx5 cells and reduces the resistance index [57]. Al-Otaibi et. al, shown that the combination of Teucrium polium L. essential oil nanoemulsion and oxaliplatin had a synergistic effect with different p53 status colon cancer cells and induced a greater percentage of apoptosis [58]. In addition, combination with dabrafenib and/or trametinib Melaleuca alternifolia (tea tree oil) synergistically reduced cell viability and prompted by apoptosis in melanoma [59].”

The work would benefit from rereading for typos and grammatical mistakes.

Author’s response: Thank you very much for your time and valuable suggestions. We are sincerely sorry for our careless mistakes. In response to your suggestions, we have revised our manuscript thoroughly and the correction has been made.

Round 2

Reviewer 2 Report

The authors have made substantial improvements in the revised manuscript. I think it is ready for publication.

The authors have made substantial improvements in the revised manuscript. I think it is ready for publication.

Author Response

Thank you!